# What Makes Freezing Layers in Deep Neural Networks Effective? A Linear Separability Perspective

**Collin Coil**[1]  **Nick Cheney**[1]

[1]Department of Computer Science & Vermont Complex Systems Institute, University of Vermont

**Abstract**  Freezing layers in deep neural networks has been shown to enhance generalization and accelerate training, yet the underlying mechanisms remain unclear. This paper investigates the impact of frozen layers from the perspective of linear separability, examining how untrained, randomly initialized layers influence feature representations and model performance. Using multilayer perceptrons trained on MNIST, CIFAR-10, and CIFAR-100, we systematically analyze the effects freezing layers and network architecture. While prior work attributes the benefits of frozen layers to Cover's theorem, which suggests that nonlinear transformations improve linear separability, we find that this explanation is insufficient. Instead, our results indicate that the observed improvements in generalization and convergence stem from other mechanisms. We hypothesize that freezing may have similar effects to other regularization techniques and that it may smooth the loss landscape to facilitate training. Furthermore, we identify key architectural factors—such as network overparameterization and use of skip connections—that modulate the effectiveness of frozen layers. These findings offer new insights into the conditions under which freezing layers can optimize deep learning performance, informing future work on neural architecture search.

## 1  Introduction

Deep neural networks, although effective for a variety of tasks, are costly to train. This is increasingly apparent as the parameter count of deep neural networks skyrocket and get trained on ever-growing datasets. In response, researchers have developed a variety of techniques to make training neural networks more efficient. One such method is by freezing layers before training, relying on the randomly initialized transformations to add expressivity to the network instead of training the entire network. Using these frozen layers with randomly initialized transformations to boost model performance is akin to using random, nonlinear high-dimension transformations in recurrent neural networks (RNNs), referred to as reservoir computing (Lukoševičius and Jaeger, 2009). Prior work on reservoir computing has focused on liquid state machines (Maass et al., 2002) and echo state networks (Jaeger, 2002). Shen et al. (2020) formally linked reservoir computing with RNNs to freezing layers, using "reservoir" as a term to describe any network with frozen layers. We adopt this terminology, referring to any network with frozen layers as reservoir networks.

We investigate freezing layers in neural networks to explain two previously documented benefits in reservoir networks: better generalization and faster training convergence, requiring fewer training steps to achieve peak performance (as demonstrated by Shen et al. (2020)). One often theorized explanation for both the better generalization and faster training convergence is through application of Cover's theorem (Cover, 1965), interpreted as a nonlinear mapping of a data points into a higher dimension feature space are more likely to be linearly separable than were the points in their original, lower dimension input space (Gallicchio and Micheli, 2023; Kanevski et al., 2002; Shen et al., 2020). Although this idea is often cited as the theoretical explanation for the improved generalizability and faster training convergence of reservoir networks demonstrated in a variety of experiments (Shen et al., 2020), we have not identified any work that has empirically validated whether increased linear separability predicted from application of Cover's theorem explains the

AutoML 2025

observed benefits of frozen layers. We explore this by investigating the linear separability of features extracted from the hidden states of fully connected neural reservoir networks.

Specifically, this paper investigates the following questions:

- Does freezing layers improve the linear separability of learned representations? We find that reservoir networks often have increased hidden state linear separability (see Section 4.1). This is more pronounced when using skip connections (see Section 4.1.3).

- How does freezing layers affect generalization and performance in reservoir networks? Freezing layers usually but not always increases generalizability (see Sections 4.1.1, 4.1.2, and 4.1.3).

- What architectural conditions influence the benefits of freezing layers? Freezing layers improves performance in massively overparameterized networks (see Sections 4.1.1, 4.1.2, and 4.1.3). Additionally, the position of the frozen layers impacts whether generalization improves (see Appendix A.2.3).

- Does Cover's theorem explain increases in generalization and convergence rate in reservoir networks? Our results indicate that Cover's theorem provides only a partial explanation, and other factors play a larger role (see Sections 4.1 and 4.2).

- What alternative explanations account for the observed effects? The effectiveness of freezing layers is more likely due other factors. We hypothesize that the benefits from freezing layers in neural networks either stems from freezing as a a form of regularization or its ability to smooth the loss landscape rather than the application of Cover's theorem. We provide intuition and preliminary evidence to support these hypotheses (see Section 4.2 and Appendix A.2.2).

## 2 Background & Related Work

**Freezing Parts of Neural Networks**. Freezing parts of neural networks is a common practice in transfer learning as a way to speed up model convergence or to reduce forgetting (Dar et al., 2022; Lee et al., 2019; Liu et al., 2021; Xiao et al., 2019). This approach is increasingly applied to randomly initialized networks for improving training efficiency. Recent work has focused on developing techniques to sequentially freeze layers during training (Brock et al., 2017; S. Li et al., 2024; Wang et al., 2023), freezing layers in varied architectures (Shen et al., 2020), freezing parts of layers (Isikdogan et al., 2020), freezing individual weights (Miao and Zhao, 2023), or using freezing layers as a dropout alternative Goutam et al., 2020. Shen et al. (2020) also explored the effect of the position of the frozen layers, finding that alternating frozen and trainable layers led to the best performance. Some research shows that networks with trainable batch normalization and all other parameters frozen can achieve high performance (Frankle, Schwab, et al., 2020). Additionally, researchers connected freezing parts of neural networks with pruning and other methods of identifying and training sparse networks (Wimmer et al., 2023). Related to pruning is the Lottery Ticket Hypothesis (LTH) (Frankle and Carbin, 2018), which states that sufficiently overparameterized networks likely contain a sparse subnetwork with equal performance. Both freezing randomly initialized layers and the LTH place importance on randomly initialized parameters. Freezing uses random layers as general feature extractors to improve performance while the LTH uses them to find optimal subnetworks. Furthermore, as we demonstrate in Section 4.1, the benefits of freezing layers in neural networks manifest in overparameterized networks, which connects to a core tenet of the LTH suggesting that neural networks are massively overparameterized. The key difference between freezing, which we investigate in this work, and pruning-based strategies (inclusive of both post-training pruning and the LTH) is that freezing keeps all weights active in the network, but some weights are not updated during training. This means that information flows through all weights of the network, even if the weight never trains. In pruning-based strategies, certain weights are replaced with 0, resulting in that connection being removed in the network.

Some prior work has discussed the impact of pruning on neural network linear separability. Lengellé and Denoeux (1996) measured linear relationships between internal representations and outputs using the sample coefficient of multiple determination (CMD), and the authors found that networks could be slightly pruned without any reduction in the CMD. Additional pruning then led to more substantial reductions in CMD. Jiang et al. (2021) illustrated that linear separability increased with pruning ratio for ResNet-18 trained on a subset of CIFAR-100 until a ratio of 90%. After that, linear separability scores fell dramatically. This pattern mirrors findings from the LTH (Frankle and Carbin, 2018): performance improves with moderate pruning but deteriorates beyond a critical threshold, suggesting a shared underlying mechanism where moderate sparsification sharpens representations impacting both linear separability and overall network accuracy, while excessive pruning degrades them.

**Linear Probing**. The goal of linear probing is to understand the inner workings of deep neural networks using simple linear classifiers. Alain and Bengio (2016) introduced linear probes to explore the dynamics of intermediate layers and diagnose network pathologies. They used a densely connected map into a softmax output layer with cross-entropy to assess linear separability. Since then, numerous researchers have used linear classifier probes to better explore neural network hidden layers in a variety of models and tasks (Fan et al., 2024; Frati et al., 2024; Xu et al., 2025; Zhang et al., 2023).

**Neural Architecture Search**. This work's investigation of neural architecture and its connection with hidden state linear separability and overall network performance is reminiscent of work in neural architecture search (NAS). In NAS, the goal is to automatically identify an architecture with optimal performance on a certain task (Elsken et al., 2019). One naive method to assess neural architectures is a grid search over a search space of architectural parameters, which has been widely discussed and applied (Liashchynskyi and Liashchynskyi, 2019; Schmitz et al., 2024). Although far more sophisticated NAS search strategies exist (Chitty-Venkata et al., 2023), basic sweeps over architectural parameters help understand what architectural features are relevant for a search space. Freezing parts of networks is not an entirely new concept in NAS. For example, Fahlman and Lebiere (1989) introduced Cascade-Correlation, which added units to a network one-by-one and froze its input weights after being added. B. Chen et al. (2021) introduced BN-NAS, which used a supernet with frozen weights and trained only the batch normalization layers of the supernet to help identify candidate subnetworks. However, the initially frozen weights were eventually unfrozen for subnet retraining. While these demonstrate that some approaches have used freezing as a tool to facilitate NAS, we have not found a NAS approach that has leaves randomly initialized and never-trained parameters in the final models.

## 3 Experimental Setup

We train multilayer perceptrons on MNIST (LeCun et al., 1998), CIFAR-10 (Krizhevsky, Hinton, et al., 2009), and CIFAR-100 (Krizhevsky, Hinton, et al., 2009) to explore the relationship between freezing layers, linear separability, and network architecture. Our networks consist of reservoir blocks, each with three layers: a trainable layer, a reservoir layer, and another trainable layer. This structure follows Shen et al. (2020), who found that alternating trainable and frozen layers optimizes performance. We explored the impact of the position of the frozen reservoir layers in the network and included the results of that investigation in Appendix A.2.3. In reservoir networks, reservoir layers remain fixed at initialization during training. In both reservoir and fully trainable networks, we scale the reservoir layer width by a factor. Our base networks have two reservoir blocks for a total of 6 layers, a reservoir layer scaling factor of two, alternating trainable and reservoir layers, and no regularization. For experiments on MNIST, we set the network's base width to 64 neurons. For experiments on CIFAR-10 or CIFAR-100, we set the network's base width to 256 neurons. For each set of hyperparameters, we train 25 reservoir networks and 25 fully trainable networks of the

same configuration for comparison. We perform a $t$-test of the accuracy scores to understand the significance of the difference between reservoir networks and fully trainable networks.

We assess the linear separability of hidden states of the neural network using a least-squares solvers trained on the features of each hidden layer using the training set. We use those least-squares solvers to evaluate linear separability of the hidden state features for both the training and testing sets. This linear separability score is the accuracy of the linear solver for classification using the hidden state features. As such, linear separability scores are constrained between 0 and 1, with 1 indicating perfect accuracy and linear separability of the hidden state features. Linear separability scores are often similar to the network's overall accuracy score; therefore, are best interpreted relative to accuracy and other linear separability scores on the same dataset. Although linear probes traditionally use a densely connected map into a softmax operation (Alain and Bengio, 2016), we find that using a least-squares solver produces a similar result with substantially less compute. In Appendix A.1, we demonstrate that linear separability scores evaluated using a least-squares solver and a traditional densely connected map are highly correlated with $r^2 = 0.978$ and $r^2 = 0.864$ for training and testing linear separability scores, respectively.

All experiments were run on an NVIDIA V100 GPU and Xeon(R) Gold 6130 CPU with TensorFlow 2.17.0 (Abadi et al., 2016). Code and results are available on GitHub. [1]

## 4 Results

### 4.1 Link between Linear Separability and Model Performance

In the following set of results, we demonstrate the relationship between neural network architecture, linear separability, and network performance. We present accuracies and figures on the average hidden state linear separability scores of the testing sets of CIFAR-10 and CIFAR-100. We provide linear separability plots using the MNIST testing set in Appendix A.2.4, training sets of each dataset in Appendix A.3, and tables of average model accuracy in Appendix A.5. We examine freezing and other regularizers in Appendix A.2.2 and the position of the reservoir layers in A.2.3.

The following sweeps over architectural parameters serve multiple purposes. First, they allow us to explore the applicability of Cover's theorem to explain the performance boost seen from freezing layers in neural networks. Sweeps over network widths and reservoir layer scaling factor allow us to investigate nonlinear random transformations in increasing dimensions. Intuitively, increasing network width and reservoir layer scaling factors results in mappings to higher dimensions, which should result in more linearly separable data. With regard to network depths, prior research has documented that linear separability increases with network depth (Alain and Bengio, 2016). As such, we expect that deeper networks display more linear separability. The hyperparameter sweeps we present in the following section provide insights for NAS on reservoir networks by exploring how freezing layers and network architecture interact to influence overall performance.

**4.1.1 Network Width.** We first examine how network width interacts with frozen layers and linear separability. Prior research suggests that freezing layers improves generalizability (Shen et al., 2020). From Cover's theorem, we expect wider networks to enhance linear separability by projecting data into higher-dimensional spaces. This sweep explores how width impacts separability, generalizability, and the capacity of frozen layers to create a set of functions to span the space. For CIFAR-10 and CIFAR-100, we assess base widths from $2^5$ to $2^{10}$ neurons. For MNIST, we assess base widths from $2^4$ to $2^9$ neurons. Each trainable layer has neurons equal to the base width, and each reservoir layer has neurons equal to the base width times a scaling factor of 2.

Figure 1 shows that wider networks consistently exhibit higher linear separability. Across all widths, reservoir networks outperform fully trainable ones in both separability (see Figure 1) and

---

[1]The following GitHub repository contains all code and results related to this paper: `https://github.com/CollinCoil/freezing-linear-separability`

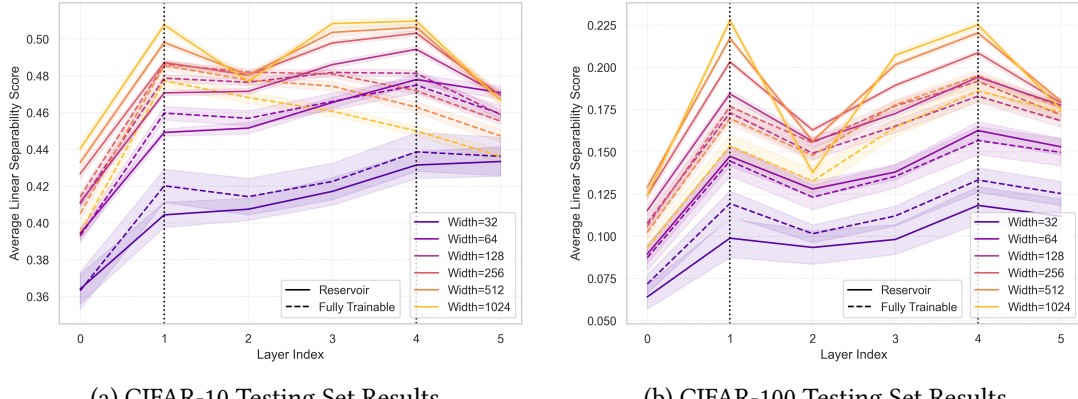

(a) CIFAR-10 Testing Set Results  (b) CIFAR-100 Testing Set Results

Figure 1: Average linear separability of the hidden state features for CIFAR-10 (a) and CIFAR-100 (b) with 95% confidence intervals for a sweep over network widths. Wider networks produced greater average hidden state feature linear separability scores. In reservoir networks, the frozen, reservoir layers are denoted by the vertical dotted black lines. All other layers are trainable in reservoir networks. In fully trainable networks, all layers are trainable, including the wider reservoir layers.

test accuracy ($p < 0.01$ for each width; see Table 1 for CIFAR-10 and Table 14 in the supplement for CIFAR-100). An exception occurs at the narrowest width (32 neurons), where fully trainable networks achieve significantly higher test accuracy and linear separability scores. This suggests overparameterization is necessary for freezing layers to provide benefits. Results for MNIST are shown in supplementary material Section A.2.4 and show that freezing layers does not improve model test accuracy for any network base width. Finally, comparing Figure 1a with Table 1 confirms a strong correlation between final-layer separability and network accuracy, reinforcing the link between linear separability and performance.

Table 1: CIFAR-10 width sweep tabular results. Reservoir is reservoir model; trainable is fully trainable model. Bold accuracy scores represent significantly greater average accuracy for the reservoir models or fully trainable models. Bold p-values signify scores less than 0.01 based on a t-test of the mean accuracy scores of the reservoir model and fully trainable model.

| CIFAR-10 | Training Accuracy | | | Testing Accuracy | | |
|---|---|---|---|---|---|---|
| Width | Reservoir | Trainable | P-value | Reservoir | Trainable | P-value |
| 32 | 0.483 | 0.501 | 1.146E-01 | 0.436 | 0.439 | 7.399E-01 |
| 64 | 0.606 | **0.638** | **1.818E-07** | **0.472** | 0.460 | **2.119E-04** |
| 128 | 0.750 | **0.796** | **1.209E-15** | **0.465** | 0.456 | **6.441E-05** |
| 256 | 0.852 | **0.882** | **7.880E-15** | **0.463** | 0.456 | **3.470E-03** |
| 512 | 0.888 | **0.900** | **1.147E-04** | **0.463** | 0.449 | **4.075E-05** |
| 1024 | 0.876 | 0.886 | 4.354E-02 | **0.464** | 0.436 | **4.436E-12** |

**4.1.2 Reservoir Layer Scaling Factor.** Intuitively, Cover's theorem suggests that if you transform data using a random nonlinear map, the data tend to be more linearly separable the transformation goes into increasingly higher dimension. We explore this by sweeping over scaling factors for the reservoir layer, which determines the number of neurons in the reservoir layer relative to the network's base width. We assess scaling factors from 0.25 to 32 in powers of 2.

As shown in Figure 2, larger reservoir layers generally yield higher separability in layers 1 and 4, except at extreme scaling factors (32 times in CIFAR-10, 16 times and 32 times in CIFAR-100),

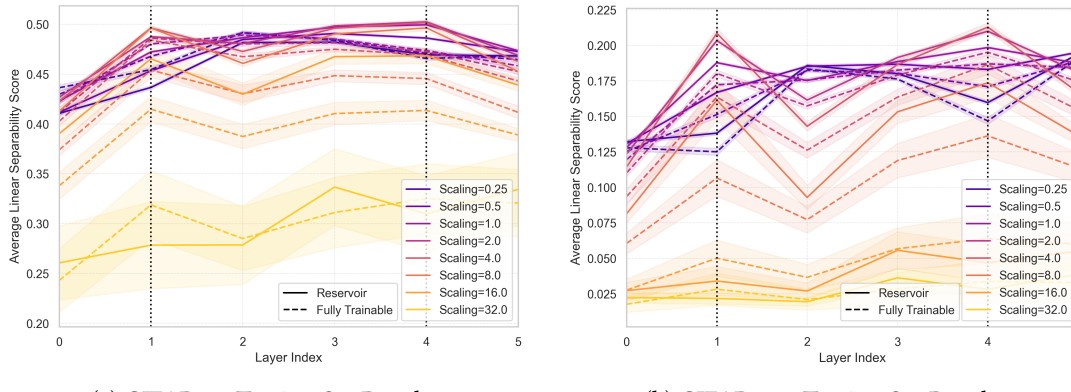

(a) CIFAR-10 Testing Set Results

(b) CIFAR-100 Testing Set Results

Figure 2: Average linear separability of the hidden state features for (a) CIFAR-10 and (b) CIFAR-100 for a sweep over reservoir layer scaling factor. Reservoir networks with greater scaling factors produced greater average linear separability in the reservoir layer (denoted by the dotted black line) The exception to this trend is networks with scaling factors of 32 in (a) and networks with scaling factors of 16 or 32 in (b). These networks often failed to train, meaning they never performed substantially better than random chance.

where networks often failed to train due to overparameterization, likely resulting in vanishing gradients. Freezing layers led to increased testing set accuracy in CIFAR-10 for most scaling factors greater than or equal to 1 ($p \ll 0.01$ for scaling factors 1-16, $p \approx 0.54$ for scaling factor of 32; see Table 2). In CIFAR-100, reservoir networks with scaling factors from 0.25 to 8 had increased testing accuracy ($p \approx 0.95$ for scaling factor of 0.25, $p \ll 0.01$ for scaling factors 0.5-8; see Table 15).

Table 2: CIFAR-10 reservoir layer scaling factor sweep tabular results. Reservoir is reservoir model; trainable is fully trainable model. Bold accuracy scores represent significantly greater average accuracy for the reservoir models or fully trainable models. Bold p-values signify scores less than 0.01.

| CIFAR-10 | Training Accuracy | | | Testing Accuracy | | |
|---|---|---|---|---|---|---|
| Scaling Factor | Reservoir | Trainable | P-value | Reservoir | Trainable | P-value |
| 0.25 | **0.861** | 0.804 | **1.022E-28** | 0.453 | **0.463** | **5.811E-07** |
| 0.5 | **0.888** | 0.842 | **6.778E-29** | 0.461 | 0.462 | 3.696E-01 |
| 1 | **0.878** | 0.868 | **5.247E-05** | **0.466** | 0.458 | **2.719E-05** |
| 2 | 0.857 | **0.877** | **5.003E-07** | **0.463** | 0.453 | **4.123E-05** |
| 4 | 0.811 | **0.866** | **4.577E-16** | **0.459** | 0.443 | **1.550E-07** |
| 8 | 0.747 | **0.803** | **1.870E-06** | **0.456** | 0.402 | **4.745E-13** |
| 16 | 0.625 | 0.633 | 7.858E-01 | **0.443** | 0.382 | **7.017E-18** |
| 32 | 0.369 | 0.369 | 9.954E-01 | 0.334 | 0.319 | 5.392E-01 |

Unexpectedly, scaling factors less than 1 led to higher separability between the first trainable layer (0) and the first reservoir layer (1) when reservoir layers were frozen, contradicting the intuitive application of Cover's theorem. This suggests trainable layers adapt to leverage frozen layers' expressivity. Nevertheless, this result calls into question the application of Cover's theorem to explain the increased performance of neural networks. In the second reservoir layer of the networks (layer index 4), we see that linear separability scores decrease from layer 3 to layer 4, aligning with intuition. This result underscores the complex interaction between network architecture and frozen layers, suggesting that Cover's theorem alone may not fully explain benefits from freezing layers.

**4.1.3 Network Depth.** A deeper network with more reservoir blocks results in more nonlinear mappings into higher dimension followed by reducing dimension and noise. This, coupled with the fact that deeper layers tend to extract more linearly separable features (Alain and Bengio, 2016), could make increasing depth a major architectural influence on hidden state feature linear separability and overall network performance. To assess this, we sweep over networks of one reservoir block deep to eight reservoir blocks deep (i.e., three to 24 hidden layers).

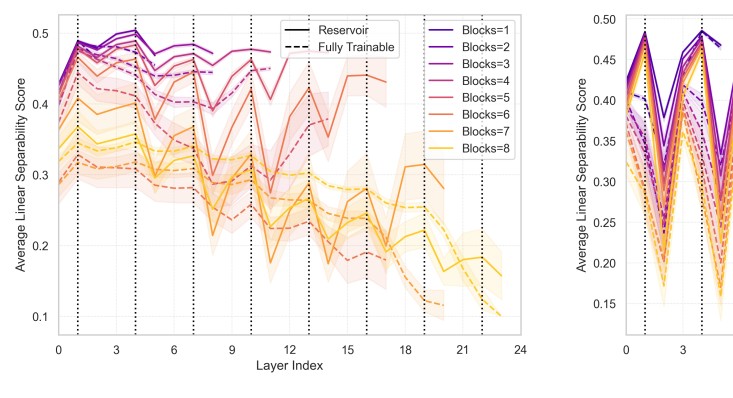

(a) CIFAR-10 Testing Set Results   (b) CIFAR-10 Testing Set Results with Skip Connections

Figure 3: Average linear separability of the hidden state features for (a) networks trained on CIFAR-10 and (b) networks with skip connections trained on CIFAR-10 for a sweep over network depth. Reservoir networks had higher average linear separability scores and accuracies for all depths on CIFAR-10 and most on CIFAR-100. Skip connections dramatically altered linear separability dynamics in hidden layers. Results on CIFAR-100 are in Appendix A.2.1.

The results of this sweep are shown in Figure 3. Freezing layers produced higher average hidden state linear separability scores and accuracies for all depths on CIFAR-10 ($p \ll 0.01$ for each depth; see 10). On CIFAR-100, networks with 6, 7, or 8 reservoir blocks failed to train, resulting in performing no better than chance. However, in networks with fewer reservoir blocks, freezing layers produced significantly higher accuracy ($p \ll 0.01$ for each depth; see Table 16). Notably, fully trainable deep networks (8 blocks in CIFAR-10, 5 or more in CIFAR-100) often failed, whereas frozen counterparts performed significantly better, highlighting layer freezing as a regularization mechanism that stabilizes training in overparameterized models. Freezing layers serves to constrain certain parameters to remain at their initial values, preventing them from updating during the optimization process. This constraint clearly has a regularization effect, often reducing overfitting on the training set and improving generalization to the testing set.

Surprisingly, linear separability does not increase monotonically with depth, contradicting prior findings (Alain and Bengio, 2016). This suggests that reservoir dynamics are somewhat contrary to prior findings. This is further supported by an investigation into the position of the frozen reservoir layers in a network. We find that the location of the frozen reservoir layers substantially impacts the representations learned throughout the network (see Appendix A.2.3). We also performed a set of experiments to compare freezing to other kinds of regularization, finding that networks with both frozen layers and traditional regularization strategies were overpenalized and did not perform as well, suggesting that the benefits of freezing and regularization may manifest through the same mechanism (see Appendix A.2.2).

Given that frozen layers enable training deep networks, we examine whether skip connections—another common deep network trick—interact with layer freezing. Inspired by Shen et al. (2020), who described freezing as a "cheap way to increase depth," we assess whether skip connections complement or duplicate this effect. We modify networks (depths 2–8 blocks) by adding skip

connections from the output of the first layer of a reservoir block to the pre-activation input of the next block (Figure 6). While traditional skip connections pass over entire blocks, we start ours following the first layer of a reservoir block to avoid needing to correct for mismatched dimensions.

The results show that in networks with skip connections, frozen layers still yield higher accuracy ($p \ll 0.01$; see Tables 13 and 19) and higher linear separability than fully trainable counterparts. The linear separability scores in networks with skip connections are radically different than networks without skip connections. In networks without skip connections, linear separability scores trended down progressing through the network. However, the addition of skip connections prevented a downward trend in linear separability scores. Moreover, skip connections alter how features evolve within reservoir blocks. In fully trainable networks, average hidden state linear separability peaks at the beginning of a reservoir block before falling in the subsequent two layers, suggesting key features propagate through the first layer while later layers contribute less. In reservoir networks, however, we see a different dynamic. While linear separability score shoots up at the beginning of each block, it peaks in the frozen layer before collapsing in the final trainable layer of the block. This suggests that while both freezing layers and skip connections serve to enhance trainability of deeper networks, they do so through different and complementary mechanisms.

### 4.2 Learned Linear Separability

The above results relate to the increased generalizability in reservoir networks but are uninformative about the faster training convergence, as illustrated by fewer training iterations necessary for reservoir networks to reach peak performance. To assess this, we explore the linear separability of hidden state features throughout the training process.

The fact that neural networks learn linearly separable features during training has been previously documented (Xu et al., 2025). However, we are unaware of an explicit demonstration that reservoir networks learn linearly separable features faster than their fully trainable counterparts. To investigate this, we perform a new experiment. In addition to freezing entire layers, we freeze a certain percentage of weights per layer, ensuring that the weights do not update during the training process. We sweep over freezing 10% to 50% of weights in each layer of the network. The goal is to explore whether the speedup in training convergence is from freezing layers or if it can more generally be achieved by freezing random weights.

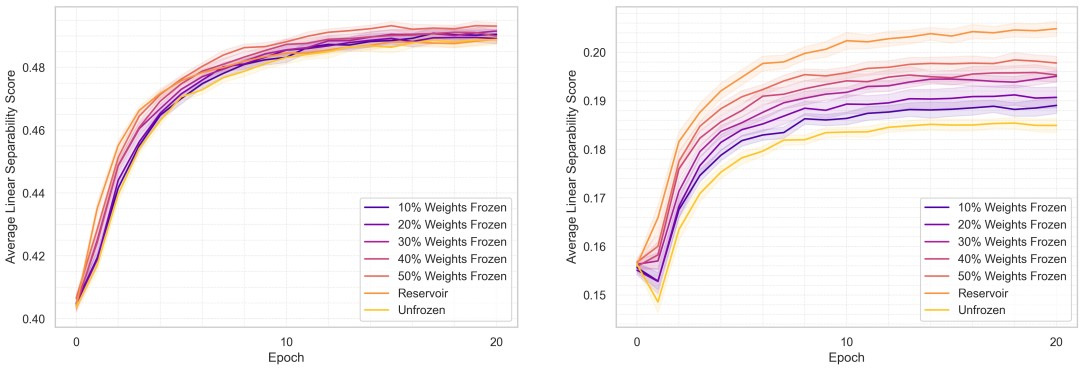

(a) CIFAR-10 Testing Set Results          (b) CIFAR-100 Testing Set Results

Figure 4: Average linear separability of the hidden state features from layer 1 for (a) CIFAR-10 and (b) CIFAR-100 over the first 10 training epochs. Networks with frozen parameters converge faster than fully trainable networks, regardless of whether random weights or entire layers are frozen. Freezing more frozen parameters results in networks learning linearly separable faster. Note that layer 1 is a reservoir layer, meaning it is completely frozen in reservoir networks. Similar results from other layers are in appendix Section A.4.

In Figure 4, we see that freezing parameters leads to networks learning linearly separable features faster than fully trainable networks. Interestingly, freezing more parameters in the network results in faster learning of linearly separable features. Since this increase in convergence speed is seen in both networks with frozen layers (i.e., reservoir networks) and randomly frozen parameters, it appears that this phenomenon is likely a result of something besides application of Cover's theorem. However, it is interesting to note that layer-frozen networks learn linearly separable features fastest, including networks with more frozen parameters (the reservoir networks have about 19% frozen parameters). These findings extend to accuracy, which is presented in Tables 3 and 20. These tables demonstrate that reservoir networks increase accuracy fastest, and increasing the percent of randomly frozen weights resulted in networks learning faster. We suspect that the increase in convergence rate is a result of freezing parameters having a smoothing effect on the loss landscape, but this remains an open question for future study.

Table 3: CIFAR-10 testing accuracy for the first several epochs. Reservoir is reservoir model; trainable is fully trainable model. Columns with percentages represent networks with that percent of weights randomly frozen per layer. Bold accuracy scores represent significantly greater average accuracy for the reservoir models or random weight frozen models when compared to the fully trainable model (p-values less than 0.01).

| CIFAR-10 | | | Random Weights Frozen | | | | |
|---|---|---|---|---|---|---|---|
| Epoch | Unfrozen | Reservoir | 10% | 20% | 30% | 40% | 50% |
| 0 | 0.101 | 0.099 | 0.101 | 0.100 | 0.101 | 0.098 | 0.100 |
| 1 | 0.347 | 0.357 | 0.353 | 0.346 | 0.356 | 0.355 | 0.353 |
| 2 | 0.386 | 0.397 | 0.391 | 0.390 | 0.393 | 0.396 | **0.398** |
| 3 | 0.412 | **0.426** | 0.414 | 0.415 | 0.416 | 0.418 | **0.424** |
| 4 | 0.429 | **0.440** | 0.432 | 0.433 | 0.435 | **0.438** | 0.438 |
| 5 | 0.444 | **0.454** | 0.445 | 0.446 | 0.451 | 0.450 | 0.449 |

## 5 Conclusion

Our investigation to understand the mechanisms powering the increased generalizability and convergence rate of reservoir networks reveals several key insights that challenge conventional understanding and offer new directions for neural architecture design.

First, Cover's theorem, which posits that nonlinear transformations increase the likelihood of linear separability, has often been cited to explain the benefits of freezing layers. However, our findings suggest that Cover's theorem alone is insufficient to account for the observed improvements, as even networks with scaling factors less than one benefited from freezing (see Section 4.1.2), contradicting the application of Cover's theorem. We suspect that both the benefits from freezing layers and the impacts on hidden state linear separability scores are caused by a third, unexplored factor (such as smoothness of the loss landscape potentially caused by the freezing parameters or the inability to overfit parameters that we do not train). This discrepancy underscores the need for alternative explanations to fully understand the advantages of freezing layers.

Second, results demonstrate a deep interplay between network architecture and the effectiveness of layer freezing. Overparameterized networks, characterized by wide base widths, high reservoir layer scaling factors, and increased depth, consistently benefited from freezing layers (see Section 4). Additionally, skip connections dramatically altered the dynamics of the linear separability scores in the network (see Section 4.1.3). This architectural dependency for the effects of freezing layers has significant implications for NAS, suggesting that the strategic freezing of layers could be a critical factor in optimizing network architecture. By understanding how architecture influences the benefits of freezing, researchers can design more efficient and effective neural networks.

Third, we provided two alternative explanations for why freezing layers causes increased generalizability and convergence speed. We suspect that the phenomena are either a result of freezing being a regularization technique or that freezing smooths the loss landscape. Our experiments showed that freezing layers often became redundant when combined with other regularization techniques, indicating an overlap in their effects (see Section A.2.2). Furthermore, freezing random weights led to similar increases in model convergence rates, suggesting that this effect is not limited to freezing entire layers but extends to freezing individual parameters (see Section 4.2). This insight points to the potential that freezing parameters may smooth the loss landscape of the networks, facilitating a more stable, direct convergence. Supporting this, prior work shows deeper networks have more chaotic loss landscapes (H. Li et al., 2018), and freezing may make networks optimize like shallower ones. This is in line with intuitions from Shen et al. (2020) which stated that freezing is a tool to facilitate training deeper networks. Future work should assess these hypotheses.

This work was limited by its focus on fully connected networks, so additional work should assess the validity of the findings for networks with other architectures (e.g., transformers and CNNs). Additionally, it was limited by the simplicity of our hyperparameter sweeps. In each experiment, we merely changed one hyperparameter at a time. Performing a grid search over multiple parameters would provide more insight on the interplay between network architecture, freezing layers, and network performance. Finally, this work focused exclusively on the linear separability of the testing set features, and Appendix A.3 demonstrates that networks have substantially different trends in the training and testing linear separability scores. Analyzing the differences between the scores may provide insights on neural network training dynamics.

Overall, this work demonstrated that freezing layers is a previously unexplored dimension of NAS that can substantially benefit networks through improved generalizability and faster training. This indicates that freezing layers may be a new direction for NAS and efficient AI as we attempt to design better networks that perform more efficiently without compromising performance.

## 6 Broader Impact Statement

Our work reiterates prior research that freezing layers in neural networks leads to networks having higher performance with faster convergence. Reservoir networks are more efficient to train, requiring fewer computations per training step and fewer epochs to train. Therefore, freezing layers serves as a technique to make deep learning more efficient, reducing the environmental impacts and financial requirements for our field. The training efficiencies from freezing layers could result in less data being required to train models, enabling wider use of deep learning models on domains with smaller curated datasets. Additionally, understanding what causes freezing leads to neural network performance improvement provides insights into how deep neural networks learn. This insight, along with the parameter sweeps demonstrating the relationship between network architecture, performance, and hidden state linear separability, can assist future researchers to design more efficient neural networks. After careful consideration, we do not see clear negative societal impacts from our work.

**Acknowledgements**. This material is based upon work supported by the National Science Foundation under Grant No. 2239691 and 2218063 and by a Presidential Doctoral Fellowship from the University of Vermont. Computations were performed on the Vermont Advanced Computing Core supported in part by NSF Award No. OAC-1827314. Additionally, we thank the reviewers for their thoughtful and specific feedback.

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

# A  Supplementary Materials

## A.1  Assessing Least-Squares versus Densely Connected Map for Linear Separability

Our approach to measuring linear separability with a least-squares solver diverges from the standard practice of using a densely connected map. However, the two produce substantially similar results. To illustrate this, we trained 25 reservoir networks on CIFAR-10, extracted hidden state features for the training and testing set, and used the features to assess linear separability scores using both a least-squares solver and densely connected map. Figure 5 shows that both solvers produce similar results. There is high correlation for the linear separability scores from the least-squares and dense probes ($r^2 = 0.978$ on the training separability scores and $r^2 = 0.864$ on the testing separability scores). While there are minor differences between the linear separability scores, we do not believe that these differences substantially alter the conclusions of this work—specifically that linear separability of hidden state features is insufficient to explain the effects of freezing layers in neural networks.

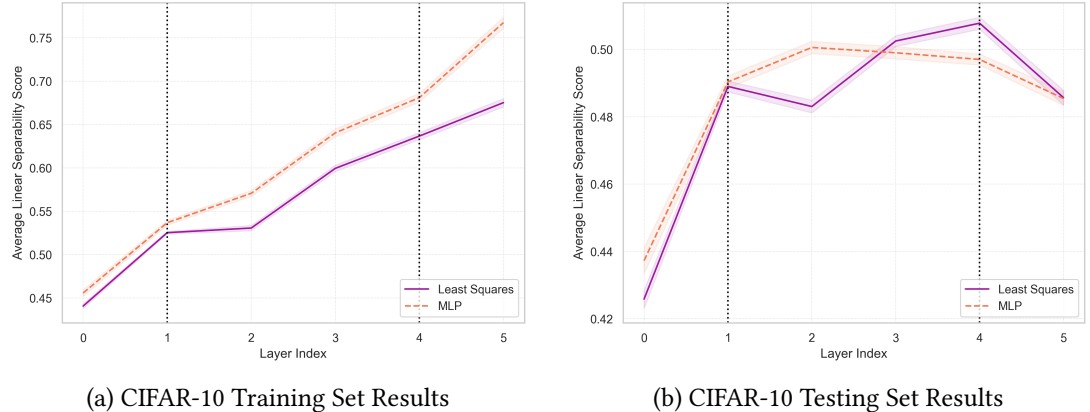

(a) CIFAR-10 Training Set Results          (b) CIFAR-10 Testing Set Results

Figure 5: Average linear separability of the CIFAR-10 (1) training set hidden state features and (b) testing set hidden state features. Linear separability scores from the least-squares solver and densely connected map resulted in similar outputs for linear separability scores.

## A.2  Testing Set Performance During Architectural Sweeps

### A.2.1  Depth and Skip Connections on CIFAR-100. Experiments on CIFAR-100 demonstrate consistent patterns in linear separability over sweeps of network depth.

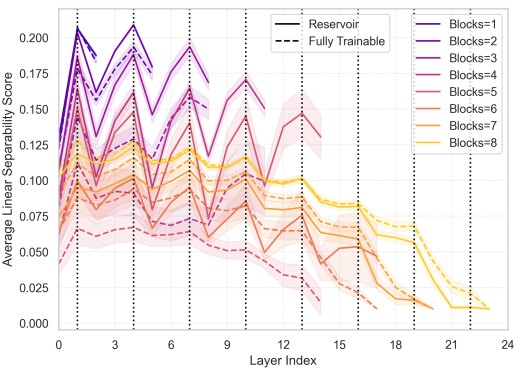

(a) CIFAR-100 Testing Set Results

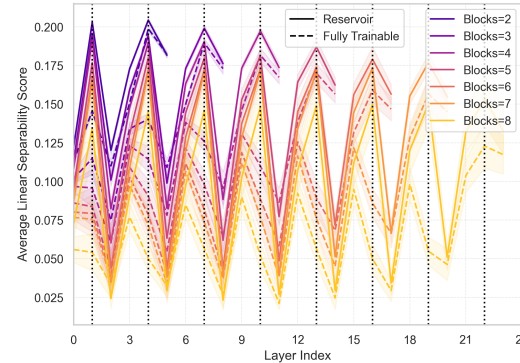

(b) CIFAR-100 Testing Set Results with Skip Connections

Figure 6: Average linear separability of the hidden state features for (a) networks trained on CIFAR-10 and (b) networks with skip connections trained on CIFAR-10 for a sweep over network depth. Reservoir networks had higher average linear separability scores and accuracies for all depths on CIFAR-10 and most on CIFAR-100. Skip connections dramatically altered linear separability dynamics in hidden layers.

**A.2.2 Regularization on CIFAR-10 and CIFAR-100.** The results in the main text suggest that freezing layers acts as some kind of implicit regularization, reducing overfitting. This raises the question of whether the effects of freezing layers will be seen in networks with conventional regularization strategies. To explore this, we applied L1 regularization, batch normalization, or dropout to all the layers in the network.

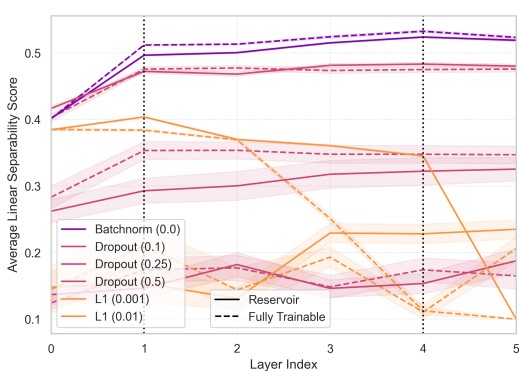

(a) CIFAR-10 Testing Set Results

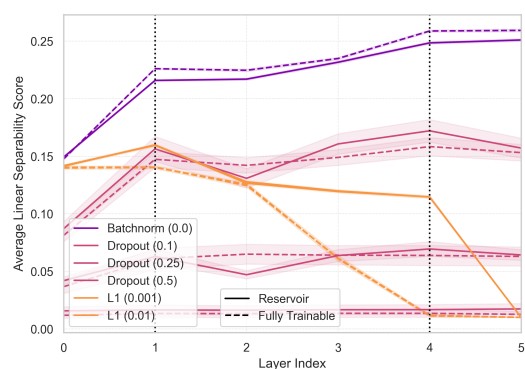

(b) CIFAR-100 Testing Set Results

Figure 7: Average linear separability of the hidden state features for (a) CIFAR-10 and (b) CIFAR-100 for a sweep over regularization strategies. Freezing layers has an inconsistent effect on overall network performance when other regularization strategies are applied.

The results of this sweep are shown in Figure 7. On CIFAR-10, reservoir networks achieved significantly higher testing accuracy than their fully trainable counterparts when using L1 regularization with a factor of 0.001 ($p \approx 0.025$; see Table 12). On CIFAR-100, reservoir networks achieved higher testing accuracy than their fully trainable counterparts when using dropout with a rate of 0.1 and 0.5, but neither was significant. Adding batch normalization to all layers or 25% dropout resulted in fully trainable networks having higher average accuracy (see Table 18). This suggests that the regularization effect seen from freezing layers sometimes is composable with the effect from other regularization strategies, but often leads to overregularization and hurts performance.

**A.2.3 Frozen Layer Position on CIFAR-10 and CIFAR-100.** While the nonlinear mapping can assist in making the data more linearly separable, the random nature of the mapping can add noise. As such, a trainable layer or layers following the reservoir layer may serve to filter out that noise (Shen et al., 2020). We explore this by rearranging the position of the trainable and reservoir layers. Our base configuration is alternating, which means that we have blocks of a trainable layer, reservoir layer, and trainable layer. We also explore putting all reservoir layers at the beginning of the network (front), stacking all reservoir layers in the middle with no trainable layers in between (middle), and placing all reservoir layers at the end of the network (back).

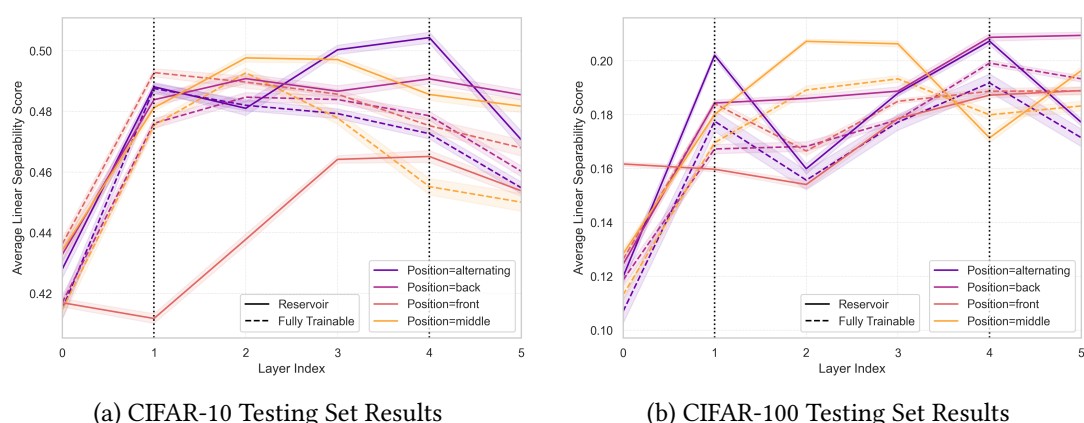

(a) CIFAR-10 Testing Set Results        (b) CIFAR-100 Testing Set Results

Figure 8: Average linear separability of the hidden state features for (a) CIFAR-10 and (b) CIFAR-100 for a sweep over frozen layer position. The location of the reservoir layers substantially impacted the linear separability scores of the hidden states. Notably, we see lower linear separability scores in networks with all reservoir layers at the front of the network.

The results of this sweep are shown in Figure 8. In both CIFAR-10 and CIFAR-100, the position of the scaled reservoir layers, whether trainable or not, substantially impacted the average linear separability scores of the networks. Additionally, we found that reservoir networks achieved higher average accuracy than their fully trainable counterparts for every position of the scaled layers with the exception of putting those layers in the front (see tables 11 and 17). Once again, we see that the impact of the position of the reservoir layers impacts network testing performance and average hidden state linear separability score, demonstrating the link between network architecture and performance.

**A.2.4 MNIST.**

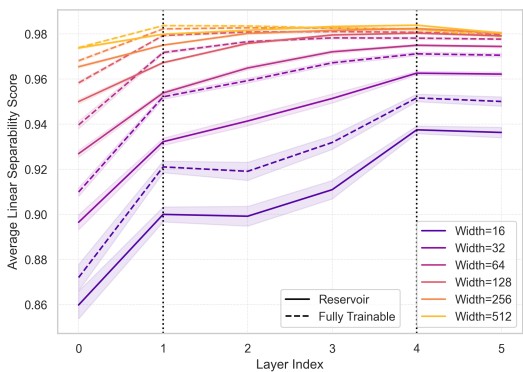

Figure 9: Average linear separability of the hidden state features for MNIST for a sweep over base widths. Similar to experiments on CIFAR-10 and CIFAR-100, wider networks produced greater average hidden state feature linear separability scores. However, freezing layers resulted in networks with lower average linear separability scores and accuracies.

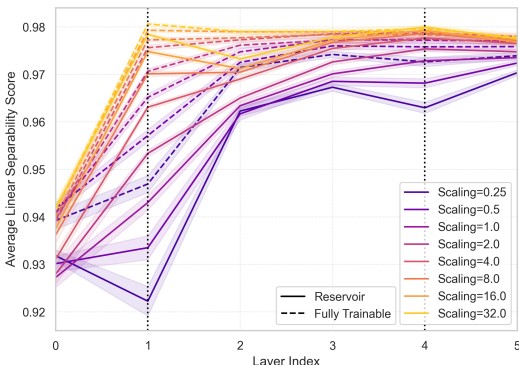

Figure 10: Average linear separability of the hidden state features for MNIST for a sweep over reservoir layer scaling factors. Similar to experiments on CIFAR-10 and CIFAR-100, greater scaling factors produced greater average hidden state feature linear separability scores in the reservoir layers (denoted by the dotted black line).

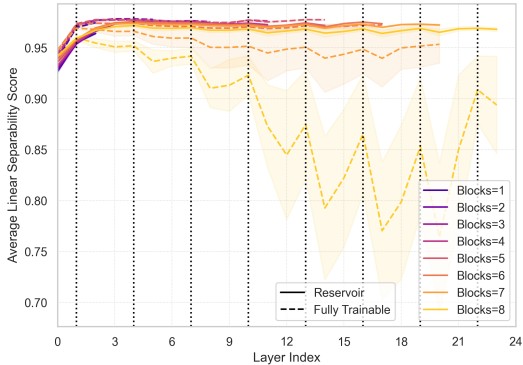

Figure 11: Average linear separability of the hidden state features for MNIST for a sweep over network depth. Similar to experiments on CIFAR-10 and CIFAR-100, freezing layers increases hidden state linear separability scores in massively overparameterized networks.

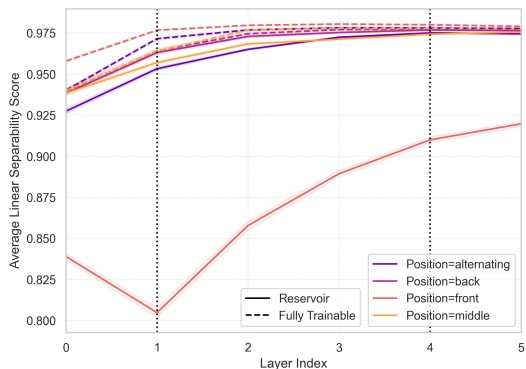

Figure 12: Average linear separability of the hidden state features for MNIST for a sweep over frozen layer position. We see that networks with both frozen layers up front have lower linear separability scores than all other networks.

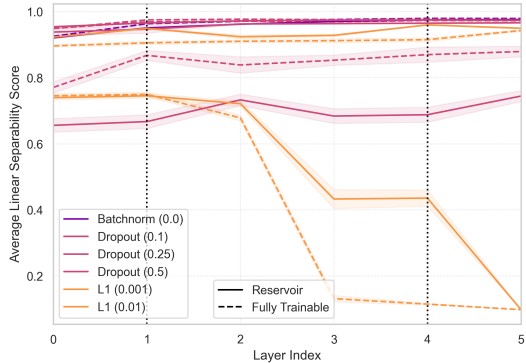

Figure 13: Average linear separability of the hidden state features for MNIST for a sweep over regularization strategies. Similar to experiments on CIFAR-10 and CIFAR-100, reservoir networks outperformed fully trainable networks with low L1 regularization.

## A.3 Training Set Performance During Architectural Sweeps

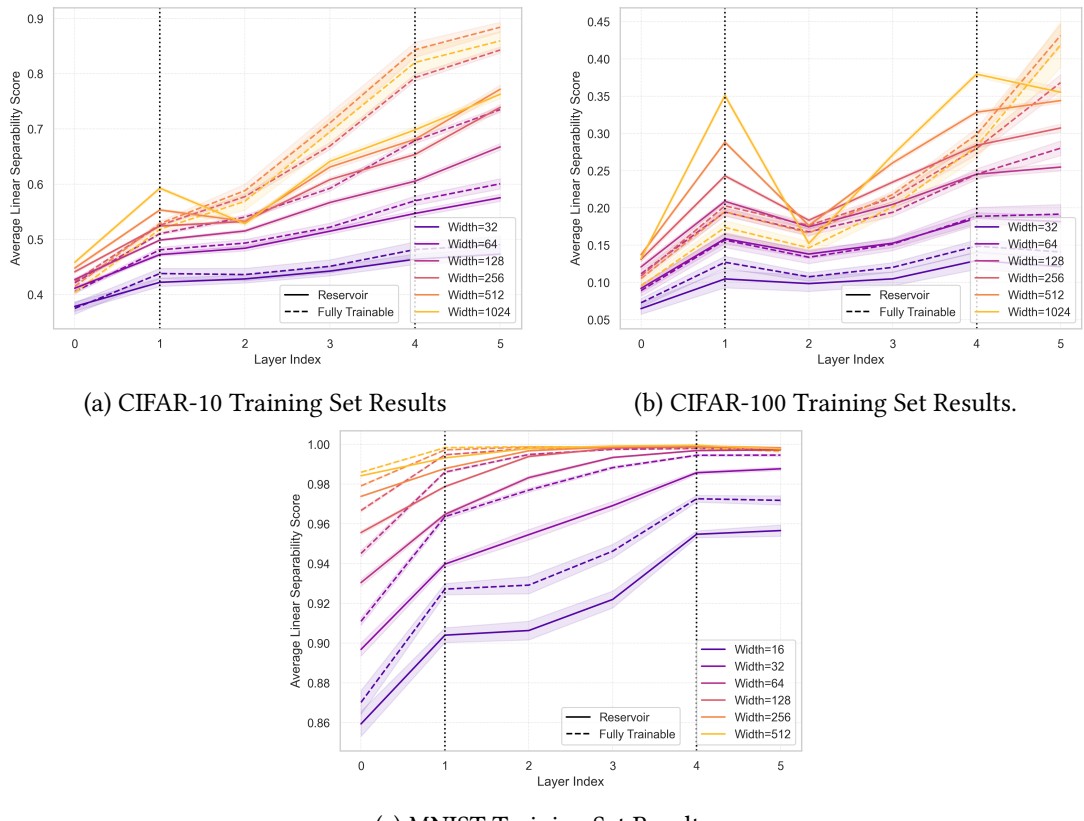

(a) CIFAR-10 Training Set Results

(b) CIFAR-100 Training Set Results.

(c) MNIST Training Set Results.

Figure 14: Average linear separability of the training set hidden state features for (a) CIFAR-10, (b) CIFAR-100, and (c) MNIST for a sweep over network widths.

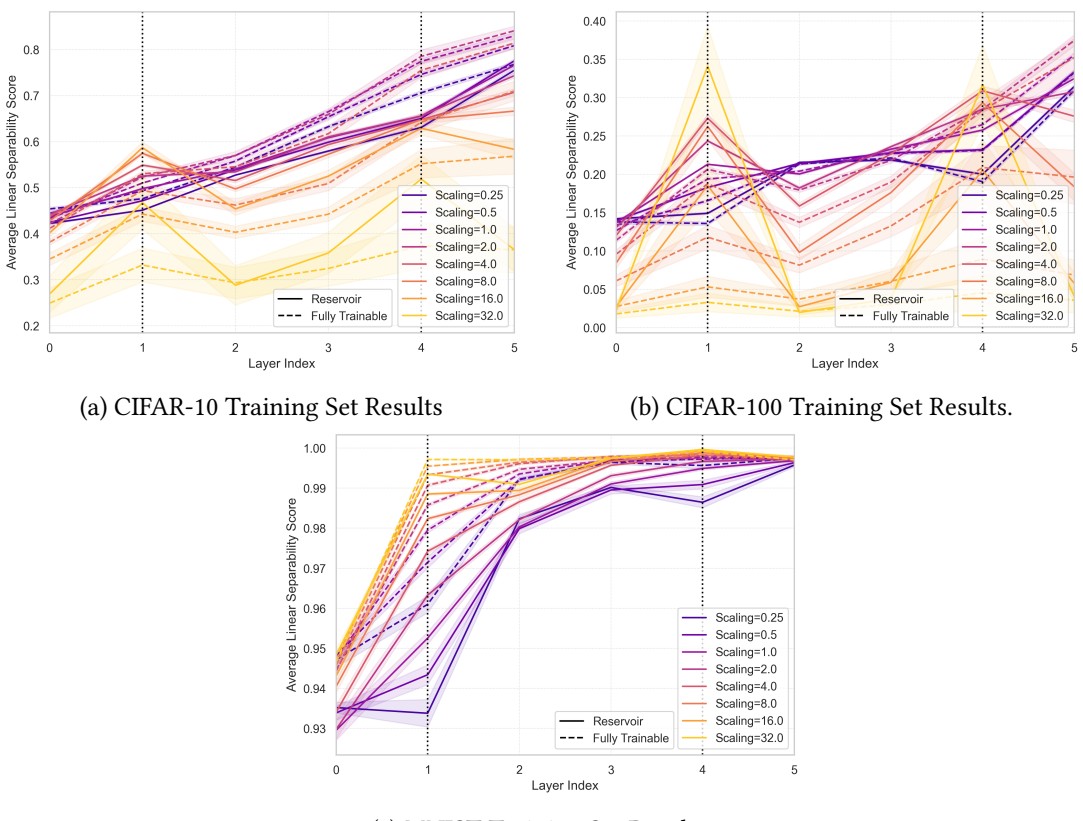

(a) CIFAR-10 Training Set Results

(b) CIFAR-100 Training Set Results.

(c) MNIST Training Set Results.

Figure 15: Average linear separability of the training set hidden state features for (a) CIFAR-10, (b) CIFAR-100, and (c) MNIST for a sweep over reservoir layer scaling factors.

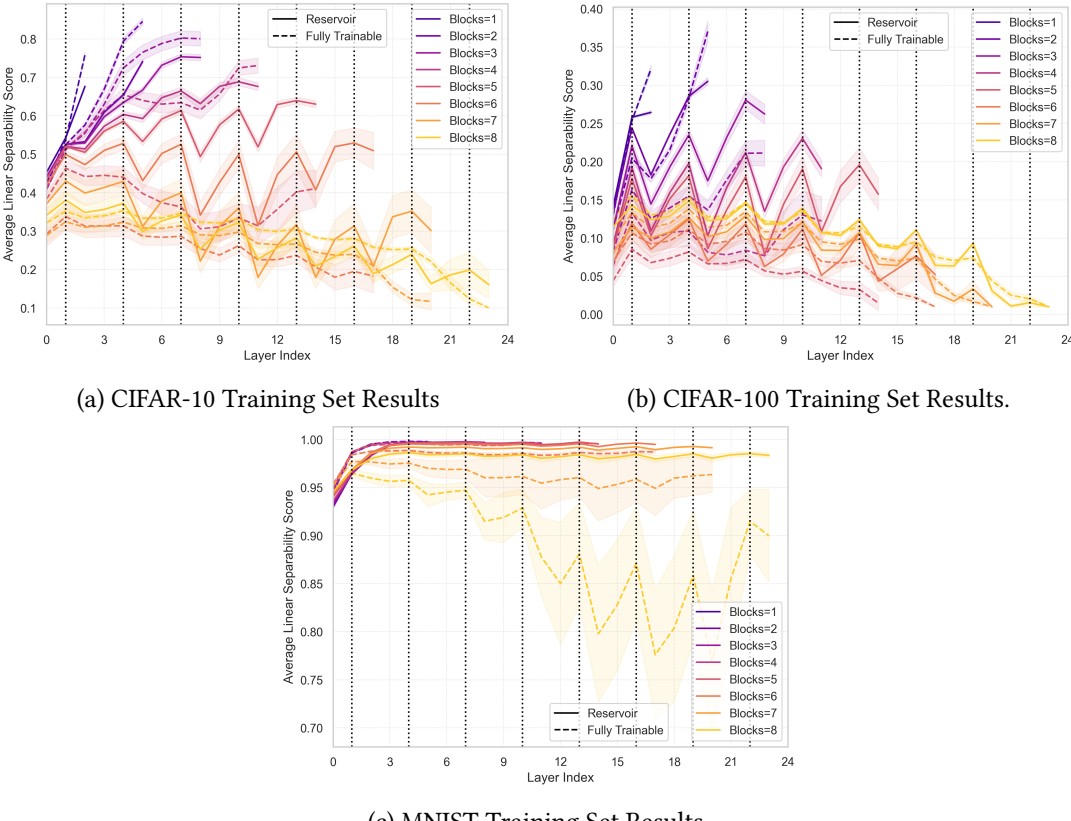

(a) CIFAR-10 Training Set Results

(b) CIFAR-100 Training Set Results.

(c) MNIST Training Set Results.

Figure 16: Average linear separability of the training set hidden state features for (a) CIFAR-10, (b) CIFAR-100, and (c) MNIST for a sweep over network depth.

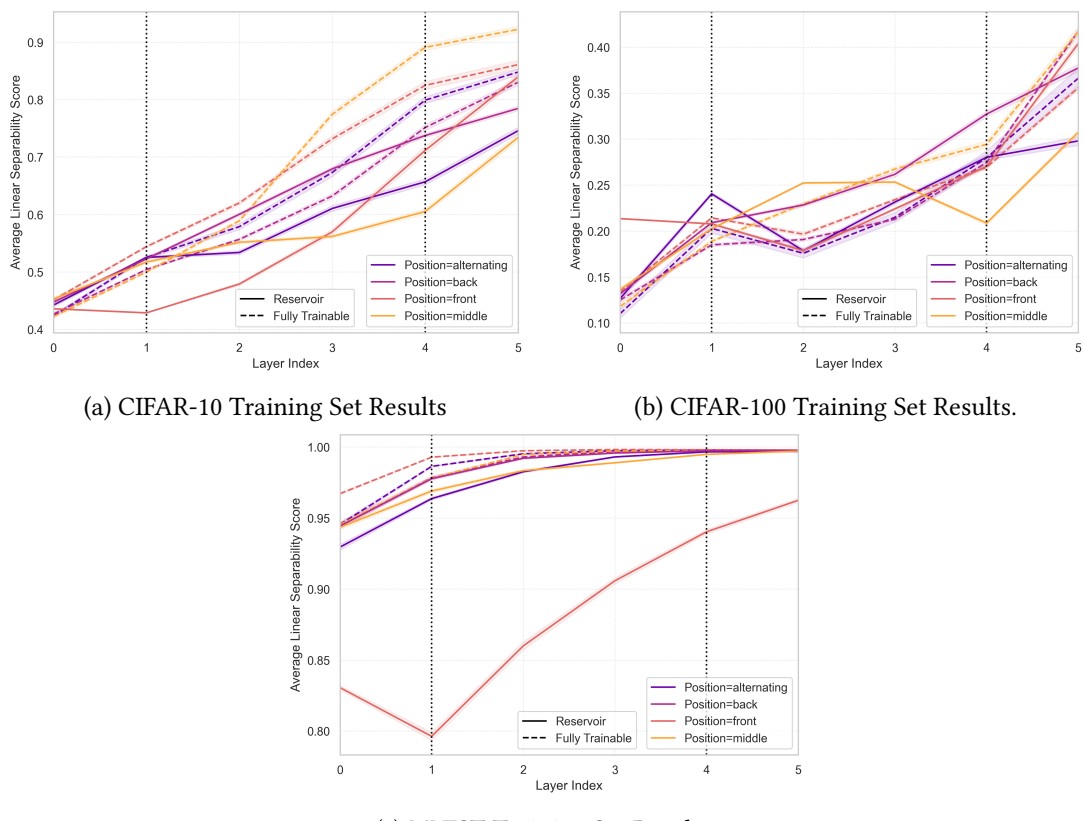

(a) CIFAR-10 Training Set Results

(b) CIFAR-100 Training Set Results.

(c) MNIST Training Set Results.

Figure 17: Average linear separability of the training set hidden state features for (a) CIFAR-10, (b) CIFAR-100, and (c) MNIST for a sweep over reservoir layer position.

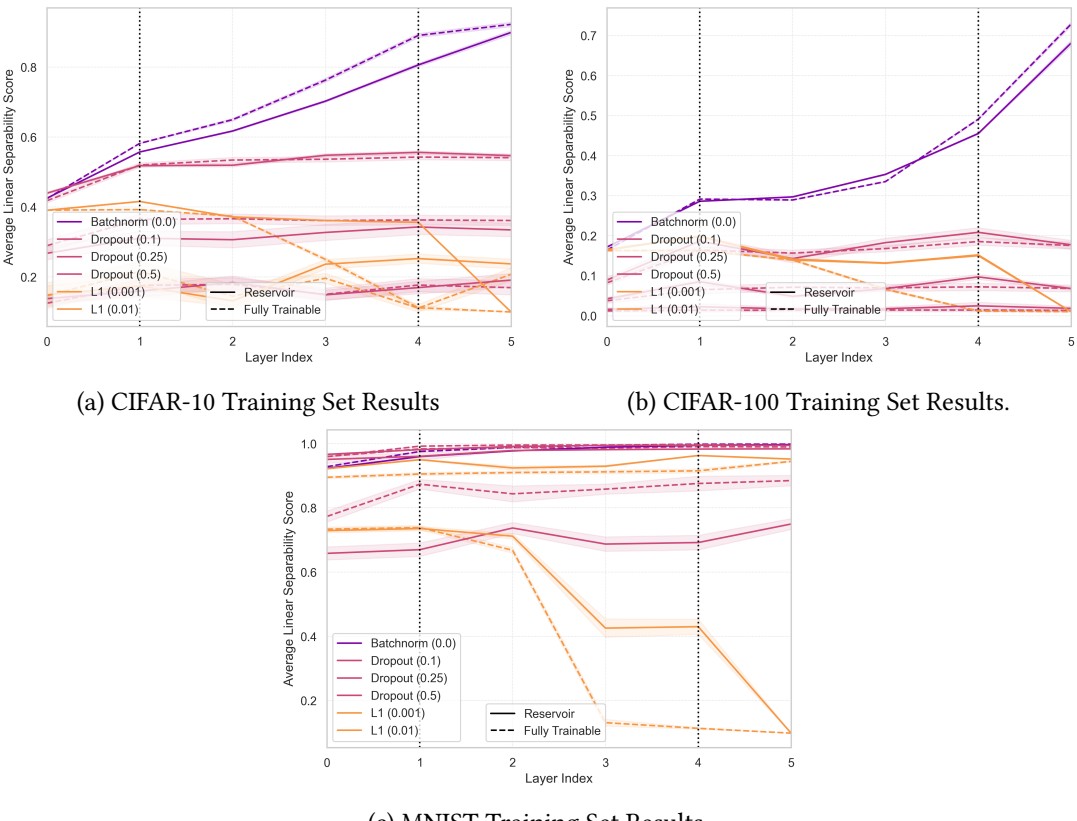

(a) CIFAR-10 Training Set Results       (b) CIFAR-100 Training Set Results.

(c) MNIST Training Set Results.

Figure 18: Average linear separability of the training set hidden state features for (a) CIFAR-10, (b) CIFAR-100, and (c) MNIST for a sweep over regularization strategies.

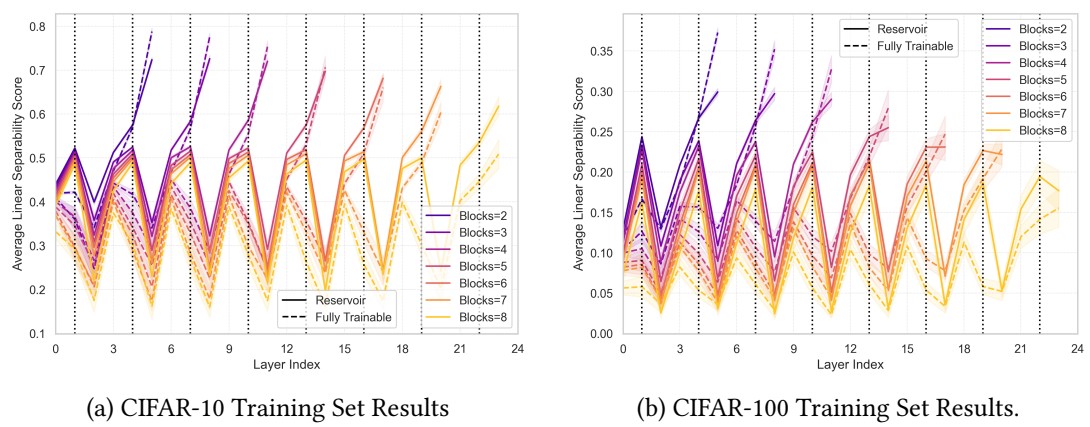

(a) CIFAR-10 Training Set Results       (b) CIFAR-100 Training Set Results.

Figure 19: Average linear separability of the hidden state features for (a) CIFAR-10 and (b) CIFAR-100 for a sweep over network depth in networks with skip connections.

## A.4 Learned Linear Separability of Networks with Frozen Parameters

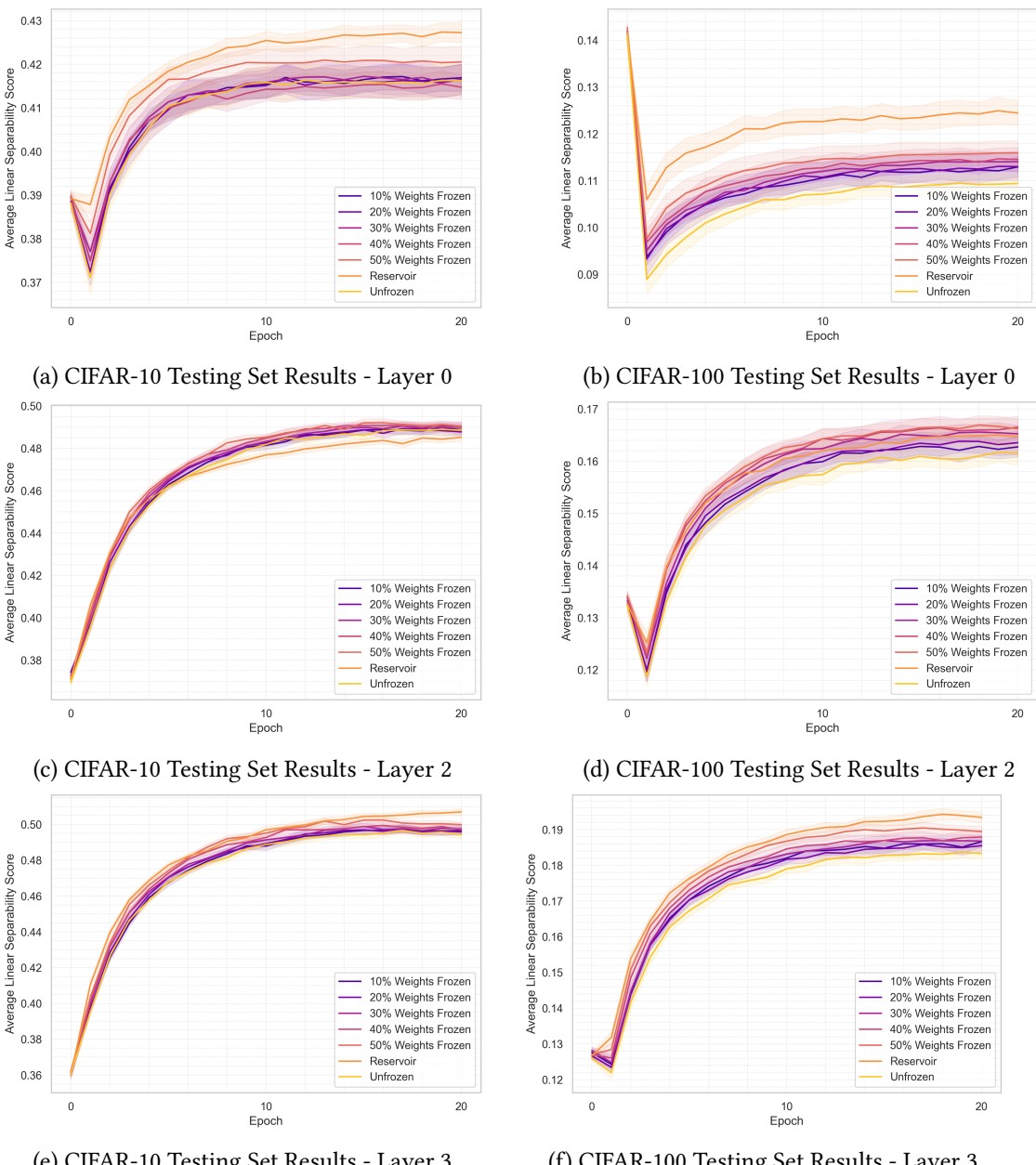

(a) CIFAR-10 Testing Set Results - Layer 0

(b) CIFAR-100 Testing Set Results - Layer 0

(c) CIFAR-10 Testing Set Results - Layer 2

(d) CIFAR-100 Testing Set Results - Layer 2

(e) CIFAR-10 Testing Set Results - Layer 3

(f) CIFAR-100 Testing Set Results - Layer 3

Figure 20: Average linear separability of the hidden state features from layers 0, 2, and 3 of networks trained on CIFAR-10 and CIFAR-100 over the first 10 training epochs. Networks with frozen parameters converge faster than fully trainable networks, regardless of whether random weights or entire layers are frozen.

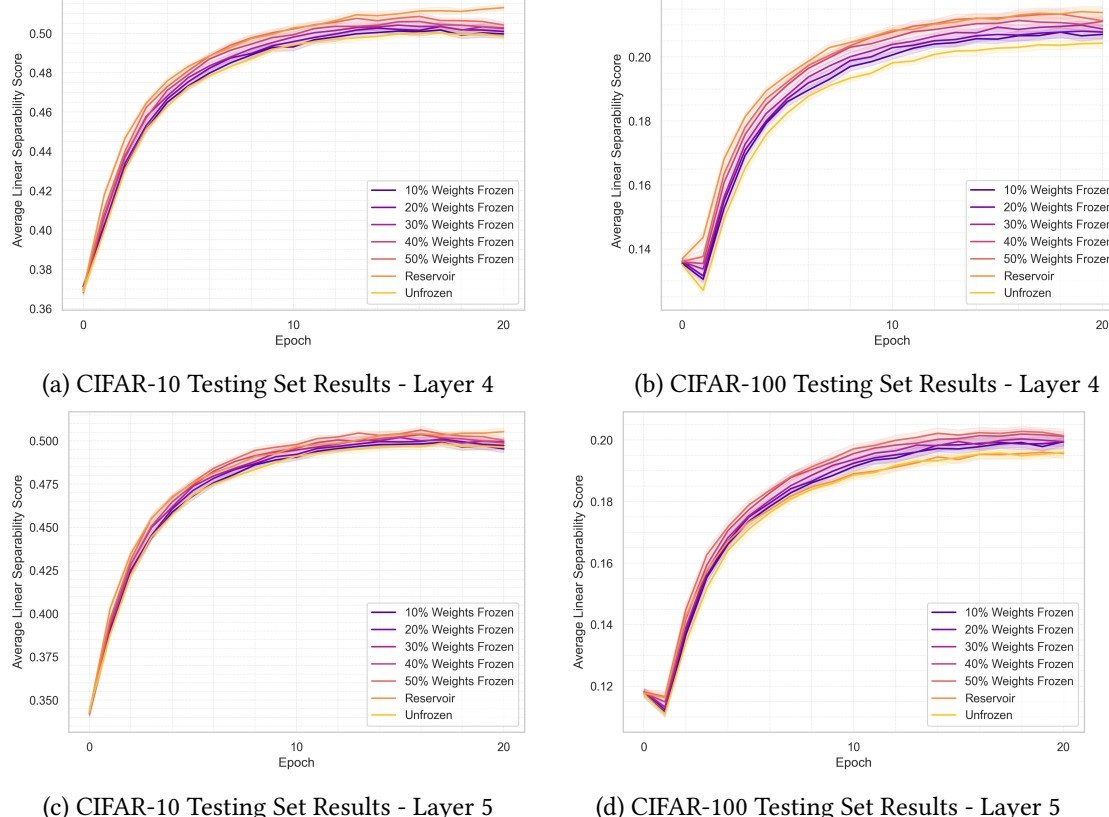

(a) CIFAR-10 Testing Set Results - Layer 4

(b) CIFAR-100 Testing Set Results - Layer 4

(c) CIFAR-10 Testing Set Results - Layer 5

(d) CIFAR-100 Testing Set Results - Layer 5

Figure 21: Average linear separability of the hidden state features from layers 4 and 5 of networks trained on CIFAR-10 and CIFAR-100 over the first 10 training epochs. Networks with frozen parameters converge faster than fully trainable networks, regardless of whether random weights or entire layers are frozen.

## A.5 Tabular Results of Average Model Accuracy

### A.5.1 MNIST.

Table 4: MNIST width sweep tabular results. Reservoir is reservoir model; trainable is fully trainable model. Bold accuracy scores represent significantly greater average accuracy for the reservoir models or fully trainable models. Bold p-values signify scores less than 0.01.

| MNIST | Training Accuracy | | | Testing Accuracy | | |
|---|---|---|---|---|---|---|
| Base Width | Reservoir | Trainable | P-value | Reservoir | Trainable | P-value |
| 32 | 0.978 | **0.983** | **3.149E-14** | 0.952 | **0.958** | **7.790E-10** |
| 64 | 0.994 | **0.995** | **2.847E-06** | 0.965 | **0.970** | **5.169E-09** |
| 128 | 0.997 | 0.997 | 1.724E-01 | 0.974 | **0.977** | **1.930E-07** |
| 256 | 0.998 | 0.998 | 7.525E-01 | 0.979 | **0.981** | **2.930E-03** |
| 512 | 0.998 | **0.998** | **3.117E-03** | 0.981 | **0.982** | **1.946E-04** |
| 1024 | 0.998 | 0.998 | 2.780E-01 | 0.982 | 0.983 | 1.490E-01 |

Table 5: MNIST reservoir layer scaling factor sweep tabular results. Reservoir is reservoir model; trainable is fully trainable model. Bold accuracy scores represent significantly greater average accuracy for the reservoir models or fully trainable models. Bold p-values signify scores less than 0.01.

| MNIST | Training Accuracy | | | Testing Accuracy | | |
|---|---|---|---|---|---|---|
| Scaling Factor | Reservoir | Trainable | P-value | Reservoir | Trainable | P-value |
| 0.25 | 0.996 | **0.997** | **3.282E-05** | 0.970 | **0.973** | **2.188E-07** |
| 0.5 | 0.997 | **0.997** | **3.689E-04** | 0.972 | **0.976** | **1.110E-07** |
| 1 | 0.997 | 0.997 | 5.867E-02 | 0.973 | **0.977** | **4.522E-07** |
| 2 | 0.997 | 0.998 | 2.912E-02 | 0.974 | **0.977** | **3.616E-03** |
| 4 | 0.997 | **0.998** | **1.382E-07** | 0.976 | **0.979** | **2.023E-07** |
| 8 | 0.997 | **0.998** | **4.316E-04** | 0.976 | **0.978** | **5.769E-05** |
| 16 | 0.997 | 0.998 | 3.510E-02 | 0.977 | **0.979** | **8.194E-03** |
| 32 | 0.997 | 0.998 | 1.103E-01 | 0.978 | **0.979** | **2.333E-03** |

Table 6: MNIST depth sweep tabular results. Reservoir is reservoir model; trainable is fully trainable model. Bold accuracy scores represent significantly greater average accuracy for the reservoir models or fully trainable models. Bold p-values signify scores less than 0.01.

| MNIST | Training Accuracy | | | Testing Accuracy | | |
|---|---|---|---|---|---|---|
| Depth (Reservoir Blocks) | Reservoir | Trainable | P-value | Reservoir | Trainable | P-value |
| 1 | 0.998 | 0.998 | 7.909E-01 | 0.973 | **0.977** | **1.293E-08** |
| 2 | 0.997 | 0.997 | 3.255E-02 | 0.975 | **0.978** | **3.415E-07** |
| 3 | 0.997 | **0.997** | **4.286E-03** | 0.974 | **0.977** | **8.189E-08** |
| 4 | 0.996 | 0.996 | 3.088E-01 | 0.974 | 0.976 | 2.531E-02 |
| 5 | **0.996** | 0.994 | **4.832E-10** | 0.974 | **0.977** | **4.106E-05** |
| 6 | **0.994** | 0.985 | **3.338E-10** | 0.974 | 0.971 | 1.938E-02 |
| 7 | **0.989** | 0.965 | **1.984E-06** | 0.972 | 0.951 | 5.950E-02 |
| 8 | **0.981** | 0.922 | **4.302E-05** | **0.968** | 0.919 | **2.165E-04** |

Table 7: MNIST reservoir layer position sweep tabular results. Reservoir is reservoir model; trainable is fully trainable model. Bold accuracy scores represent significantly greater average accuracy for the reservoir models or fully trainable models. Bold p-values signify scores less than 0.01.

| MNIST | Training Accuracy | | | Testing Accuracy | | |
|---|---|---|---|---|---|---|
| Reservoir Layer Position | Reservoir | Trainable | P-value | Reservoir | Trainable | P-value |
| Alternating | 0.997 | **0.998** | **4.112E-03** | 0.974 | **0.978** | **6.077E-09** |
| Front | 0.981 | **0.998** | **9.157E-28** | 0.924 | **0.980** | **2.163E-34** |
| Middle | 0.997 | 0.997 | 7.219E-01 | 0.975 | **0.977** | **1.778E-04** |
| Back | 0.997 | 0.997 | 2.605E-02 | 0.976 | 0.977 | 1.015E-02 |

Table 8: MNIST regularization sweep tabular results. Reservoir is reservoir model; trainable is fully trainable model. Bold accuracy scores represent significantly greater average accuracy for the reservoir models or fully trainable models. Bold p-values signify scores less than 0.01.

| MNIST | Training Accuracy | | | Testing Accuracy | | |
|---|---|---|---|---|---|---|
| Regularization | Reservoir | Trainable | P-value | Reservoir | Trainable | P-value |
| L1 0.001 | **0.956** | 0.954 | **1.311E-04** | 0.954 | 0.953 | 4.780E-01 |
| L1 0.01 | 0.112 | 0.112 | 1.000E+00 | 0.113 | 0.113 | 1.000E+00 |
| Batchnorm | 0.995 | **0.996** | **4.249E-15** | 0.978 | **0.980** | **1.586E-07** |
| Dropout 0.1 | 0.983 | **0.989** | **4.577E-36** | 0.975 | **0.978** | **5.871E-11** |
| Dropout 0.25 | 0.959 | **0.972** | **1.173E-30** | 0.968 | **0.974** | **3.119E-19** |
| Dropout 0.5 | 0.662 | **0.816** | **9.892E-18** | 0.711 | **0.872** | **1.694E-17** |

Table 9: MNIST first several epochs tabular results. Reservoir is reservoir model; trainable is fully trainable model. Bold accuracy scores represent significantly greater average accuracy for the reservoir models or fully trainable models. Bold p-values signify scores less than 0.01.

| MNIST | Training Accuracy | | | Testing Accuracy | | |
|---|---|---|---|---|---|---|
| Epoch | Reservoir | Trainable | P-value | Reservoir | Trainable | P-value |
| 1 | 0.894 | **0.901** | **2.035E-10** | 0.948 | 0.951 | 2.029E-02 |
| 2 | 0.957 | **0.959** | **8.917E-08** | 0.959 | **0.962** | **9.208E-03** |
| 3 | 0.967 | **0.969** | **2.154E-07** | 0.964 | 0.966 | 6.028E-02 |
| 4 | 0.973 | **0.975** | **1.188E-08** | 0.967 | 0.969 | 1.074E-01 |
| 5 | 0.977 | **0.978** | **2.974E-09** | 0.968 | **0.971** | **5.091E-05** |

## A.5.2 CIFAR-10.

Table 10: CIFAR-10 depth sweep tabular results. Reservoir is reservoir model; trainable is fully trainable model. Bold accuracy scores represent significantly greater average accuracy for the reservoir models or fully trainable models. Bold p-values signify scores less than 0.01.

| CIFAR-10 | Training Accuracy | | | Testing Accuracy | | |
|---|---|---|---|---|---|---|
| Depth (Reservoir Blocks) | Reservoir | Trainable | P-value | Reservoir | Trainable | P-value |
| 1 | 0.782 | **0.909** | **2.941E-28** | **0.476** | 0.451 | **1.353E-18** |
| 2 | 0.857 | **0.879** | **1.424E-15** | **0.464** | 0.455 | **7.309E-06** |
| 3 | 0.768 | **0.797** | **4.935E-03** | **0.472** | 0.446 | **8.091E-15** |
| 4 | 0.668 | **0.711** | **4.126E-04** | **0.474** | 0.449 | **1.035E-17** |
| 5 | **0.617** | 0.415 | **1.175E-09** | **0.469** | 0.386 | **4.461E-05** |
| 6 | **0.507** | 0.179 | **3.123E-14** | **0.434** | 0.176 | **5.526E-13** |
| 7 | **0.317** | 0.116 | **4.193E-07** | **0.303** | 0.118 | **2.632E-07** |
| 8 | **0.173** | 0.098 | **1.868E-03** | **0.171** | 0.100 | **1.843E-03** |

Table 11: CIFAR-10 reservoir layer position sweep tabular results. Reservoir is reservoir model; trainable is fully trainable model. Bold accuracy scores represent significantly greater average accuracy for the reservoir models or fully trainable models. Bold p-values signify scores less than 0.01.

| CIFAR-10 | Training Accuracy | | | Testing Accuracy | | |
|---|---|---|---|---|---|---|
| Reservoir Layer Position | Reservoir | Trainable | P-value | Reservoir | Trainable | P-value |
| Alternating | 0.859 | **0.882** | **7.995E-13** | **0.467** | 0.455 | **1.628E-06** |
| Front | **0.950** | 0.882 | **1.247E-37** | 0.438 | **0.469** | **6.799E-22** |
| Middle | 0.839 | **0.933** | **9.698E-27** | **0.464** | 0.454 | **8.770E-07** |
| Back | **0.878** | 0.868 | **2.085E-07** | 0.459 | 0.456 | 5.490E-02 |

Table 12: CIFAR-10 regularization sweep tabular results. Reservoir is reservoir model; trainable is fully trainable model. Bold accuracy scores represent significantly greater average accuracy for the reservoir models or fully trainable models. Bold p-values signify scores less than 0.01.

| CIFAR-10 | Training Accuracy | | | Testing Accuracy | | |
|---|---|---|---|---|---|---|
| Regularization | Reservoir | Trainable | P-value | Reservoir | Trainable | P-value |
| L1 0.001 | 0.228 | 0.201 | 2.121E-02 | 0.226 | 0.201 | 2.512E-02 |
| L1 0.01 | 0.098 | 0.098 | 6.939E-01 | 0.100 | 0.100 | 1.000E+00 |
| Batchnorm | 0.943 | 0.943 | 7.892E-01 | 0.508 | **0.515** | **1.877E-03** |
| Dropout 0.1 | 0.482 | 0.475 | 7.225E-02 | 0.471 | 0.468 | 2.761E-01 |
| Dropout 0.25 | 0.274 | 0.298 | 1.304E-02 | 0.287 | **0.323** | **6.703E-03** |
| Dropout 0.5 | 0.151 | 0.141 | 1.866E-01 | 0.161 | 0.155 | 4.805E-01 |

Table 13: CIFAR-10 depth sweep tabular results for networks with skip connections. Reservoir is reservoir model; trainable is fully trainable model. Bold accuracy scores represent significantly greater average accuracy for the reservoir models or fully trainable models. Bold p-values signify scores less than 0.01.

| CIFAR-10, Skip Conn. | Training Accuracy | | | Testing Accuracy | | |
|---|---|---|---|---|---|---|
| Depth (Reservoir Blocks) | Reservoir | Trainable | P-value | Reservoir | Trainable | P-value |
| 2 | 0.843 | **0.909** | **3.100E-22** | **0.464** | 0.454 | **9.423E-06** |
| 3 | 0.839 | **0.903** | **2.301E-19** | **0.461** | 0.447 | **2.544E-07** |
| 4 | 0.827 | **0.880** | **1.639E-11** | **0.458** | 0.437 | **8.970E-08** |
| 5 | 0.800 | **0.840** | **5.661E-03** | **0.449** | 0.422 | **4.781E-09** |
| 6 | 0.783 | 0.785 | 9.384E-01 | **0.443** | 0.408 | **1.067E-09** |
| 7 | 0.756 | 0.719 | 5.560E-02 | **0.440** | 0.395 | **4.875E-18** |
| 8 | **0.697** | 0.571 | **1.560E-04** | **0.431** | 0.383 | **3.741E-11** |

### A.5.3 CIFAR-100.

Table 14: CIFAR-100 width sweep tabular results. Reservoir is reservoir model; trainable is fully trainable model. Bold accuracy scores represent significantly greater average accuracy for the reservoir models or fully trainable models. Bold p-values signify scores less than 0.01.

| CIFAR-100 | Training Accuracy | | | Testing Accuracy | | |
|---|---|---|---|---|---|---|
| Width | Reservoir | Trainable | P-value | Reservoir | Trainable | P-value |
| 32 | 0.151 | **0.182** | **7.645E-03** | 0.134 | 0.151 | 4.863E-02 |
| 64 | 0.238 | 0.264 | 5.916E-02 | 0.178 | 0.168 | 1.161E-01 |
| 128 | 0.384 | **0.468** | **1.701E-09** | **0.194** | 0.174 | **8.521E-11** |
| 256 | 0.552 | **0.684** | **1.689E-16** | **0.183** | 0.164 | **1.803E-09** |
| 512 | 0.671 | **0.807** | **2.734E-28** | **0.180** | 0.158 | **1.450E-11** |
| 1024 | 0.711 | **0.782** | **8.216E-03** | **0.178** | 0.145 | **2.678E-10** |

Table 15: CIFAR-100 reservoir layer scaling factor sweep tabular results. Reservoir is reservoir model; trainable is fully trainable model. Bold accuracy scores represent significantly greater average accuracy for the reservoir models or fully trainable models. Bold p-values signify scores less than 0.01.

| CIFAR-100 | Training Accuracy | | | Testing Accuracy | | |
|---|---|---|---|---|---|---|
| Scaling Factor | Reservoir | Trainable | P-value | Reservoir | Trainable | P-value |
| 0.25 | **0.580** | 0.486 | **3.123E-30** | 0.180 | 0.180 | 9.489E-01 |
| 0.5 | **0.638** | 0.562 | **5.158E-20** | **0.181** | 0.176 | **4.787E-05** |
| 1 | 0.611 | **0.633** | **7.999E-05** | **0.182** | 0.174 | **1.031E-10** |
| 2 | 0.547 | **0.697** | **1.072E-25** | **0.184** | 0.165 | **2.676E-13** |
| 4 | 0.458 | **0.690** | **9.763E-10** | **0.182** | 0.146 | **1.080E-10** |
| 8 | 0.263 | 0.343 | 1.039E-01 | **0.154** | 0.110 | **7.723E-09** |
| 16 | 0.059 | 0.080 | 2.248E-01 | 0.056 | 0.062 | 5.955E-01 |
| 32 | 0.041 | 0.039 | 8.455E-01 | 0.041 | 0.033 | 3.422E-01 |

Table 16: CIFAR-100 depth sweep tabular results. Reservoir is reservoir model; trainable is fully trainable model. Bold accuracy scores represent significantly greater average accuracy for the reservoir models or fully trainable models. Bold p-values signify scores less than 0.01.

| CIFAR-100 | Training Accuracy | | | Testing Accuracy | | |
|---|---|---|---|---|---|---|
| Depth (Reservoir Blocks) | Reservoir | Trainable | P-value | Reservoir | Trainable | P-value |
| 1 | 0.430 | **0.690** | **2.203E-33** | **0.209** | 0.181 | **1.844E-28** |
| 2 | 0.546 | **0.686** | **2.394E-12** | **0.184** | 0.165 | **3.698E-09** |
| 3 | **0.398** | 0.276 | **6.776E-08** | **0.175** | 0.147 | **1.336E-07** |
| 4 | **0.232** | 0.129 | **6.319E-06** | **0.161** | 0.099 | **4.503E-05** |
| 5 | **0.174** | 0.014 | **4.560E-14** | **0.140** | 0.014 | **3.013E-15** |
| 6 | **0.054** | 0.009 | **4.578E-03** | 0.010 | 0.010 | 1.000E+00 |
| 7 | 0.009 | 0.009 | 5.625E-01 | 0.010 | 0.010 | 1.000E+00 |
| 8 | 0.009 | 0.009 | 4.022E-01 | 0.010 | 0.010 | 1.000E+00 |

Table 17: CIFAR-100 reservoir layer position sweep tabular results. Reservoir is reservoir model; trainable is fully trainable model. Bold accuracy scores represent significantly greater average accuracy for the reservoir models or fully trainable models. Bold p-values signify scores less than 0.01.

| CIFAR-100 | Training Accuracy | | | Testing Accuracy | | |
|---|---|---|---|---|---|---|
| Reservoir Layer Position | Reservoir | Trainable | P-value | Reservoir | Trainable | P-value |
| Alternating | 0.528 | **0.689** | **6.065E-22** | **0.183** | 0.161 | **1.090E-11** |
| Front | **0.844** | 0.612 | **2.866E-47** | 0.169 | **0.178** | **1.199E-11** |
| Middle | 0.541 | **0.716** | **2.483E-37** | **0.187** | 0.171 | **7.034E-15** |
| Back | 0.672 | **0.771** | **1.167E-35** | **0.178** | 0.169 | **3.451E-10** |

Table 18: CIFAR-100 regularization sweep tabular results. Reservoir is reservoir model; trainable is fully trainable model. Bold accuracy scores represent significantly greater average accuracy for the reservoir models or fully trainable models. Bold p-values signify scores less than 0.01.

| CIFAR-100 | Training Accuracy | | | Testing Accuracy | | |
|---|---|---|---|---|---|---|
| Regularization | Reservoir | Trainable | P-value | Reservoir | Trainable | P-value |
| L1 0.001 | 0.009 | 0.009 | 4.750E-01 | 0.010 | 0.010 | 1.000E+00 |
| L1 0.01 | 0.009 | 0.009 | 8.553E-01 | 0.010 | 0.010 | 1.000E+00 |
| Batchnorm | 0.835 | **0.857** | **3.664E-28** | 0.235 | **0.243** | **1.534E-08** |
| Dropout 0.1 | 0.141 | 0.137 | 5.456E-01 | 0.155 | 0.147 | 2.277E-01 |
| Dropout 0.25 | 0.041 | 0.042 | 6.936E-01 | 0.037 | 0.043 | 1.257E-01 |
| Dropout 0.5 | 0.012 | 0.010 | 2.361E-01 | 0.013 | 0.012 | 3.591E-01 |

Table 19: CIFAR-100 depth sweep tabular results for networks with skip connections. Reservoir is reservoir model; trainable is fully trainable model. Bold accuracy scores represent significantly greater average accuracy for the reservoir models or fully trainable models. Bold p-values signify scores less than 0.01.

| CIFAR-100, Skip Conn. | Training Accuracy | | | Testing Accuracy | | |
|---|---|---|---|---|---|---|
| Depth (Reservoir Blocks) | Reservoir | Trainable | P-value | Reservoir | Trainable | P-value |
| 2 | 0.532 | **0.744** | **7.794E-33** | **0.192** | 0.178 | **2.996E-12** |
| 3 | 0.528 | **0.693** | **2.815E-17** | **0.184** | 0.167 | **9.366E-09** |
| 4 | 0.504 | **0.642** | **1.955E-08** | **0.180** | 0.161 | **2.851E-07** |
| 5 | 0.421 | **0.531** | **1.974E-03** | **0.171** | 0.149 | **5.262E-07** |
| 6 | 0.362 | **0.454** | **8.994E-03** | **0.165** | 0.140 | **1.282E-05** |
| 7 | 0.342 | 0.402 | 5.477E-02 | **0.162** | 0.141 | **6.473E-04** |
| 8 | 0.254 | 0.226 | 4.419E-01 | **0.142** | 0.113 | **4.983E-03** |

Table 20: CIFAR-100 testing accuracy for the first several epochs. Reservoir is reservoir model; trainable is fully trainable model. Columns with percentages represent networks with that percent of weights randomly frozen per layer. Bold accuracy scores represent significantly greater average accuracy for the reservoir models random weight frozen models when compared to the fully trainable model (p-values less than 0.01).

| CIFAR-100 Epoch | Trainable | Reservoir | Random Weights Frozen | | | | |
| --- | --- | --- | --- | --- | --- | --- | --- |
| | | | 10% | 20% | 30% | 40% | 50% |
| 0 | 0.009 | 0.010 | 0.010 | 0.009 | 0.010 | 0.010 | 0.009 |
| 1 | 0.073 | **0.081** | 0.075 | 0.076 | 0.077 | **0.081** | **0.082** |
| 2 | 0.107 | **0.117** | 0.107 | 0.111 | 0.111 | **0.113** | **0.116** |
| 3 | 0.130 | **0.144** | 0.134 | 0.133 | 0.135 | **0.140** | **0.142** |
| 4 | 0.148 | **0.160** | 0.150 | 0.151 | **0.154** | **0.155** | **0.159** |
| 5 | 0.159 | **0.170** | 0.164 | **0.164** | **0.165** | **0.168** | **0.170** |

