# OpenReview forum: "What Makes Freezing Layers in Deep Neural Networks Effective? A Linear Separability Perspective"
_automl.cc/AutoML/2025/Methods_Track — AutoML 2025 Methods Track_

### Official Review · Reviewer_1xqa · 2025-04-29

**Comments To Authors:**

## Paper Summary
The paper investigates the influence of freezing layers which can be beneficial to make training larger networks more efficient and is thus quite relevant to NAS. Specifically, different architectural choices are evaluated combined with frozen layers and thus the paper provides a broad overview of the different effects frozen layers illicit in training.

## Decision Summary
I think this is a well done and insightful paper. I do not think the selection of datasets and architecture are a bit issue for the results' relevance and the amount of experiments they enable is probably quite important for the amount of insights that are generated here. I think this is a good paper and should be accepted.

## Strengths
I believe the overall research direction is quite strong. The effects of frozen layers are important to developing efficient NAS strategies for bigger models and the investigation focus on explaining the benefits through linear separability contextualizes the results nicely. Obviously MNIST and CIFAR with the architecture used in the paper are not the most relevant or challenging problems, but for the purpose of the study they work well enough.

The variations chosen (width, depth, reservoir scaling) represent the most important architectural changes and are explored fairly well with six to eight variations per experiment. Some of the more obvious additional questions a reader may have, e.g. about regularization or the placement of the frozen layers, are included in the supplement. These ablations offer, in my opinion, a quite good overview of how frozen layers affect different networks.

The experimental setup seems documented well, even though a table summarizing hyperparameters would be a good addition (as well as how these hyperparameters were arrived at). Nevertheless, the comparison seems sound for all datasets with a sufficient number of repetitions per experiment. The font size of the result plots could be increased a bit, though overall they are quite intuitive to read.
As some of the accuracy results tables are in the appendix, 4.1.3 and 4.2 are a bit harder to read than the previous sections, though I appreciate this is due to space constraints. Still, adding the corresponding tables to the main paper should be a priority in my opinion.

The argumentation about how the results only partially confirm Cover's theorem and in places suggest other mechanisms at work is convincing to me. Especially the results on skip connections and randomly frozen weights are interesting and insightful. The discussion of potential reasons for the effects seen throughout the experiments was interesting to me, as are the suggestions for future work. Overall, I enjoyed reading this paper and believe it will be a great entry point to the topic of freezing layers.

## Weaknesses

The paper's biggest weakness is probably the model selection since especially CNNs would likely be more relevant to the current literature. As stated above, I do not think this invalidates the claims or the paper's insights and I would not argue for rejection on these grounds. Apart from this, I only have small criticisms related to information that I could not immediately find in the paper:

- I am not sure whether I overlooked the results that the least square solver achieves comparable results to the map to softmax approach, but I would expect the authors include at least the preliminary results that led to this claim. I do not doubt it is true, but I would assume the empirical evidence for any claim stated as "we find" can be found in the paper if only for completeness' sake.

- Similary, I had trouble finding what the documented p-values were for. A T-test across frozen and fully trainable runs? This should be included in the main text (and if it is at the moment and I only overlooked it, perhaps referring to it a bit more prominently would be good).

I already mentioned the font size of the plots, I really think it should be increased. There's also a bracket not closed in line 211. But overall, these are minor points.

**Review Confidence:**

2

**Review Rating:**

9

---

### Official Review · Reviewer_c3aR · 2025-05-01

**Comments To Authors:**

**Summary**:
This paper investigates how freezing random layers impact the generalizability and training convergence of linear networks. It explores the linear separability of extracted features and questions whether freezing layers improves the linear separability of learned representations. The study also examines the architectural conditions that influence these benefits. This study is conducted on linear networks with different widths and depths on MNIST, CIFAR-10, and CIFAR-100.


**Strengths**:
This paper is clearly written and presents an interesting study of what drives the improved performance of networks with frozen layers. The concept of freezing as a form of regularisation is a novel finding.

**Weaknesses**:
- As mentioned in the related work section, this study goes into the area of the Lottery-Ticket Hypothesis and potential pruning usage. However,  the differences between this study and the existing related work is not totally clear to me. How could pruning affect the linear separability and improve the performance?

- It’s worth noting that only linear networks were used, which is an important first step. However, further evaluations on more recent networks would enhance the relevance of this study for current research. But this is a point, which could be addressed in future work.

**Review Confidence:**

3

**Review Rating:**

7

---

### Official Review · Reviewer_3Nrr · 2025-05-02

**Comments To Authors:**

This is an interesting paper. Refreshing to see work that is about trying to improve our understanding rather than "better performance".

I'm not sure the paper really answers the question posed in the title, but it certainly gives experimental results that expend our knowledge of the effects of layer freezing on generalization and convergence. It is clearly-written and code is available.

Specific comments:

- You set up the "reservoir network" terminology on page 1 but don't use it for quite a while.
- Sec 2, line 64 typo "Researchers are increasing applying..."
- Fig.1 and onwards: you don't actually state (e.g. the equation) to measure linear separability, but I assume is it between 0 and 1. In this case, you interpret results mostly in relative terms, but in absolute terms it seems that separability is often relatively low (i.e. 0.5 and lower). It would be worth clarifying and commenting on this.
- You say "...failed to train..." several times, but how did you decide this quantitatively? Similarly, "...training convergence..."
- Sec.5. The mention of loss landscape smoothing, even in the abstract, is perhaps overstated for the paper, since you really only have a hypothesis about this - your results don't seem to directly give us evidence.
- pg9, line 288 typo, "...and we it is..."

**Review Confidence:**

5

**Review Rating:**

9

---

### Meta-Review · Area_Chair_h9oy · 2025-05-08

**Recommendation:** Accept
**Confidence:** 4

**Metareview:**

The paper analyzes the effects of freezing randomly selected layers of a linear neural network on generalization and training efficiency.
Reviewer 1xqa highlighted the paper's relevance to neural architecture search.
Overall, the reviewers appreciated the interesting insights provided.
However, some concerns were raised about the limited focus on linear neural networks and the unclear generalizability of the conclusions to other architectures.
Given the overall positive feedback from the reviewers, I recommend acceptance of the paper.